# Long COVID at Different Altitudes: A Countrywide Epidemiological Analysis

**DOI:** 10.3390/ijerph192214673

**Published:** 2022-11-08

**Authors:** Juan S. Izquierdo-Condoy, Raul Fernandez-Naranjo, Eduardo Vasconez-González, Simone Cordovez, Andrea Tello-De-la-Torre, Clara Paz, Karen Delgado-Moreira, Sarah Carrington, Ginés Viscor, Esteban Ortiz-Prado

**Affiliations:** 1One Health Research Group, Faculty of Health Science, Universidad de Las Americas, Quito 170137, Ecuador; 2Health Management and Research Area, Department of Health Sciences, Universidad Internacional Iberoamericana, Arecibo, PR 00613, USA; 3Lugar, Medio y Sociedad Research Group, School of Economics, Universidad de Las Américas, Quito 170124, Ecuador; 4Departament de Biologia Cel·lular, Fisiologia i Immunologia, Universitat de Barcelona, 08028 Barcelona, Spain

**Keywords:** COVID-19, SARS-CoV-2, long-COVID, sequalae, symptoms, Latin America, high altitude

## Abstract

Background: Several reports from around the world have reported that some patients who have recovered from COVID-19 have experienced a range of persistent or new clinical symptoms after a SARS-CoV-2 infection. These symptoms can last from weeks to months, impacting everyday functioning to a significant number of patients. Methods: A cross-sectional analysis based on an online, self-reporting questionnaire was conducted in Ecuador from April to July 2022. Participants were invited by social media, radio, and TV to voluntarily participate in our study. A total of 2103 surveys were included in this study. We compared socio-demographic variables and long-term persisting symptoms at low (<2500 m) and high altitude (>2500 m). Results: Overall, 1100 (52.3%) responders claimed to have Long-COVID symptoms after SARS-CoV-2 infection. Most of these were reported by women (64.0%); the most affected group was young adults between 21 to 40 years (68.5%), and most long-haulers were mestizos (91.6%). We found that high altitude residents were more likely to report persisting symptoms (71.7%) versus those living at lower altitudes (29.3%). The most common symptoms were fatigue or tiredness (8.4%), hair loss (5.1%) and difficulty concentrating (5.0%). The highest proportion of symptoms was observed in the group that received less than 2 doses. Conclusions: This is the first study describing post-COVID symptoms’ persistence in low and high-altitude residents. Our findings demonstrate that women, especially those aging between 21–40, are more likely to describe Long-COVID. We also found that living at a high altitude was associated with higher reports of mood changes, tachycardia, decreased libido, insomnia, and palpitations compared to lowlanders. Finally, we found a greater risk to report Long-COVID symptoms among women, those with previous comorbidities and those who had a severer acute SARS-CoV-2 infection.

## 1. Introduction

COVID-19 irrupted into our lives in a very rapid way after the SARS-CoV-2 virus jumped from animals housed and sold in the Huanan Seafood Market in Wuhan, China, the most likely epicenter for the pandemic [1]. The rapid spread of the virus caught us off guard, resulted in significant morbidity and mortality across the world [2,3]. According to World Health Organization data, from the beginning of the pandemic until 27 September 2022, a cumulative total of 612 million cases of COVID-19, and 6.5 million deaths were reported. Currently, COVID-19 cases have been shown to have decreased in all regions (range: −33%, −7.8% decrease); COVID-19 deaths were also shown to have decreased worldwide (range: −33.9%, −3% decrease). In the Americas region, in the last week of September 2022, 486,224 cases and 3751 deaths were reported, a −14.4% fall in cases and −12.2% decrease in deaths compared with previous weeks [4].

It is estimated that approximately 80% of people who contracted this infection previous to the arrival of vaccines had a mild to moderate illness, 15% required hospitalization and 5% developed a critical illness requiring intensive care unit (ICU) [5,6]. The symptoms that lingered and persisted after contracting COVID-19 were something that was not well known at the beginning of the pandemic [7]. Perego was the first to use the term Long-COVID on social media to denote the persistence of symptoms weeks or months after the initial infection [7]. In 2021 the World Health Organization (WHO) defined Long-COVID as a “Post COVID-19 condition that occurs in individuals with a history of probable or confirmed SARS CoV-2 infection, usually 3 months from the onset of COVID-19 with symptoms that last for at least 2 months and cannot be explained by an alternative diagnosis” [8,9,10].

Seeßle et al. observed that at 12 months, only 22.9% of the patients were completely asymptomatic. It has also been recorded that at 30 days of infection, 68% of patients had at least 1 symptoms and at 60 days, 66% of patients persisted with symptoms [11,12]. Among the most frequent symptoms of Long-COVID are reduced exercise capacity (56.3%), fatigue (53.1%), dyspnea (37.5%), problems with concentration (39.6%), problems finding words (32.3%), and moderate to severe sleep disturbances (26.0%) [11].

Several studies conclude that the risk factors for developing Long-COVID are being women, the severity of the infection, admission to the ICU, age between 0–60 years, and having comorbidities such as high blood pressure or diabetes mellitus [13,14,15,16,17]. It has also been identified that having a body mass index (BMI) greater than 30 predisposes patients to present severe forms of breathlessness after recovery from the initial infection [18]. Patients receiving supplemental oxygen more frequently present cognitive impairment or neurological deficits after being discharged from COVID-19 infection [19]. Certain ethnic groups were also more affected by certain sequelae. For example, Asians were more prone to intracranial hemorrhage problems, African Americans were more affected by parkinsonism, while Caucasians reported more cases of dementia and insomnia [20].

In Ecuador, as of October 2022, 1003,778 confirmed cases and 35,894 deaths due to COVID-19 have been reported. Ecuador recorded a new wave of infections that started at the end of May 2022, the positivity rate in June (30.6%) almost doubled that recorded in May (17.2%) and exceeded more than three times that of April (8.8%) (3.4). It is believed that the stagnation of the vaccination plan is the main factor associated with this increase; now, 87.0% of the population in Ecuador has completed the two-dose COVID-19 vaccine schedule [21]. However, 6.6% are partially vaccinated with one or two boosters. Although several descriptive studies on Long-COVID have been developed in several countries around the world, the behavior of Long-COVID in populations with high rates of mestizos such as the Ecuadorian population has not been explored so far. Likewise, within the study of the physiopathology of Long-COVID, the role of hypobaric hypoxia to which high-altitude residents are exposed has not been addressed [22,23]. In this regard, it has been speculated that living at high altitudes is associated with lower mortality due to COVID-19 [24]. We believe that the phenotypic and adaptive differences that populations with a high degree of mestizos possess may cause differences in the characteristics that Long-COVID has been shown to have in previous studies.

In this context, the objective of this study is to describe the epidemiology, possible risk factors, and the main symptoms associated with Long-COVID, including a novel low and high-altitude analysis among Ecuadorian residents. 

## 2. Materials and Methods

### 2.1. Study Design and Sample Selection

We conducted a cross-sectional study by circulating an independent 36 item, self-reporting, online questionnaire through the internet-based survey free access platform “Shyni”. We gathered anonymous responses from all over the country using a non-probability sampling method from April to July 2022. Participants were invited by social media, radio, and TV to voluntarily participate in our study.

### 2.2. Settings

The study was carried out in Ecuador, one of the smallest Latin American countries, located in the equatorial line and bordering the Pacific Ocean. Ecuador shares borders with Peru and Colombia, and its current population is estimated to be 17,577,116 inhabitants [24].

### 2.3. Population

Participants residing in Ecuador during the study period, who agreed to participate voluntarily and who reported having a personal history of COVID-19 infection were included. While responses from participants who claimed to reside outside of Ecuador and who denied having a personal history of COVID-19 infection prior to completing the survey were excluded. 

With a confidence level of 99% and a margin of error of 3%, our minimum estimated sample was 1849 responses. The sample size was calculated by using the following formula:x = Z(c/100)2r(100 − r)
n = N x/((N − 1)E2 + x)
E = Sqrt[(N − n)x/n(N − 1)]
where “N” is the population size, “r” is the minimum response rate set by default in 50%, “Z(c/100)” is the critical value for the confidence level “c”, “x” is the expected population, and Sqrt is the square root. All responses included in the study came from respondents who voluntarily agreed to participate in the study and who completed all 36 questions.

The study variables analyzed were age, geographical distribution at the provincial level, sex, high altitude living, comorbidities, smoking habits, number of SARS-CoV-2 infections the person has had, vaccination status and number of doses, and Long-COVID after recovery from COVID-19 infection. In addition, we classified the severity of the infection, based on the report of treatment received, into three categories: mild infection (those who did not receive medication or self-medicated or received medication prescribed by a doctor; moderate infection (those who had to be hospitalized for less than 3 days or 3 to 7 days) and serious infection (those who were hospitalized for more than 7 days or who were admitted to the ICU).

### 2.4. Survey Development and Measures

The data was collected using a 36-item online questionnaire to evaluate self-reported Long COVID symptoms. The participant’s consent was obtained at the beginning of the questionnaire with an explanation of the objective of the study. Participants could proceed with the full questionnaire only after giving their consent by accepting (electronically marking) the Terms and Conditions and a Participation Agreement consent form. The questionnaire was developed and fielded in Spanish and later translated into English for reporting purposes. The full survey instrument is available in the Appendix A. The questionnaire was reviewed for validity by three experts in infectious diseases and biostatistics to identify key issues that may be relevant to Long-COVID symptoms and to assess its overall relevance and accuracy. In the survey, we included the EuroQol five-dimensions–3-level (EQ5D-3L) instrument that is a versatile scale to measure quality of life (QOL) with five dimensions (mobility, self-care, usual activities, pain/discomfort, and anxiety/depression) and a visual analog scale (EQ VAS) to evaluate the perceived health state of the participant [25]. After incorporating expert feedback, we pilot-tested the survey instrument online with a group of 30 eligible participants. The 30 participants who completed the pilot-testing did not participate in the final survey and the responses collected during pilot-testing were not included in the final analysis.

### 2.5. Data Management

To ensure the highest possible accuracy in our results, we review the data case by case. This process was undertaken to identify cases where the answers did not match the questions asked. This included examples where the respondent answered that they had not presented Long-COVID symptoms yet in the Long-COVID symptoms section they responded with having had symptoms. Such cases were automatically eliminated from the studied sample. Additionally, some questions were also designed to identify potential pretentious answers. Where pretentious answers were identified, respondents’ reports were also eliminated from the final sample. The data for this article was collected over the previous three months using a web interface questionnaire. No IP, or any other sensitive data was recorded from the respondents (Figure 1).

### 2.6. Statistical Analysis

Descriptive and inferential analysis was conducted using the software IBM SPSS Statistics for Windows Version 24.0 (IBM Company, Chicago, IL, USA). The results of each item in the questionnaire were reported as men and women in percentages and absolute frequencies with no further intersex variability analysis. For quantitative variables, the Kolmogórov–Smirnov test was applied to determine data distribution, and median and interquartile ranges (IQR) were used as descriptive statistics for non-parametric variables. The Chi-Square test was used to test the association between nominal qualitative variables. A Chi-Square test for the independence of variables was chosen to search for a relationship between variables such as gender, age, comorbidities, history of COVID-19, altitude exposure, smoking habits, vaccination status and number of vaccines given. To analyze the scores of the EQ-5D-3l, the index value was calculated using Ecuador’s specific scores [26]. The method for calculating the *p*-value was based on the frequentist approach to the test. Statistical significance was accepted at *p*  <  0.05. Confidence intervals at 95% from means and proportions were also computed.

### 2.7. Reliability and Validation

Reliability was examined using a test–retest questionnaire using the final version of the survey. Since this questionnaire was created only for this project, we tested within the cohort of experts previously selected for the informal interviews.

### 2.8. Ethical Approval

All procedures performed in our study were in accordance with the ethical standards of the Minister of Public Health and with the Helsinki Declaration and comparable ethical standards. The current analysis did not include any personal (name, ID number, telephone number, e-mail or home address) or sensible information, complying with all local and international guidelines regarding the ethical use of this type of information. This project is part of our research program on COVID-19, a nationwide study of the epidemiology of COVID-19 in Ecuador, which received an exemption letter from the UDLA’s CEISH in 2020 and received a granted letter from the Minister of Public Health.

## 3. Results

### 3.1. General Demographic Information

A total of 2271 responses were collected, however, after data revalidation, 2103 responses from patients with a previous history of COVID-19 infection were included. Overall, 1100 (52.3%) responders claimed to have Long-COVID symptoms. From the positive responses, 61.0% (*n* = 1282) were women and 67.4% (1418) of the respondents had between 21 and 40 years of age (Table 1).

### 3.2. Participants with Long-Term COVID19 Symptoms

#### 3.2.1. Demographic and Past Medical History 

From the total of the participants with Long-COVID (*n* = 1100), mestizos were mostly (91.6%). In terms of sex, 64% (*n* = 704) were women and from the entire cohort of patients with Long-COVID symptoms, 71.7% (*n* = 789) live within the highlands. From the Long-COVID group (*n* = 1100), 22.5% (*n* = 247) had comorbidities, being the most common hypertension (25.1%) and overweight (19.0%). Only 22.5% had previous history of smoking, while 72.9% had history of consuming alcohol (Appendix A).

Most participants used medication prescribed by a physician (73.6%) to treat acute COVID-19 infection and received two doses and a booster of the COVID-19 vaccine (71.0%). From those who had reported Long-COVID symptoms, 51.0% were infected before receiving the complete vaccination scheme. On the other hand, most symptoms occurred at the same time of infection (40.8%), and of those that experienced persistent symptoms, 30.1% said they occurred on a daily basis (Appendix A).

#### 3.2.2. Symptomatologic Analysis

Of the total number of reports (*n* = 1100), we identified 7746 events grouped into 52 different symptoms. Overall, the most self-reported symptoms were fatigue or tiredness 652 (8.4%), hair loss 391 (5.1%) and difficulty concentrating 387 (5.0%) (Appendix A). 

According to sex, most of the most frequent symptoms were similar for men and women. The only differences found were in hair loss (6% in women, and 3% in men). While the most frequent symptoms reported by men but not by women were chest pain, tachycardia, and loss of muscle strength. On the other hand, the most frequent symptoms only in women were taste dysfunction and easy crying (Figure 2).

Women reported 5405 (69.8%) of the Long-COVID symptoms. From the total number of reports (*n* = 7746), nauseas (85.3%), hair loss (80.8%) and altered sense of taste (71.9%), were found statistically higher in women (*p* < 0.05) (Appendix A). 

Regarding the ethnicity of the participants, 90.2% (*n* = 6984) of the reported LC symptoms belonged to the mestizo ethnic group, similarly, 15 symptoms affecting various parts of the body were shown to be significantly higher in the mestizo responders compared to the rest of the ethnicities. These differences could be due to sample defects (Appendix A).

#### 3.2.3. Age Group Analysis 

The distribution of symptoms between age groups showed that most symptoms were concentrated in the 21 to 30 (33.3%) and 31 to 40 years (34.5%) age groups. The most reported Long-COVID symptoms among participants aged 21 to 30 years were taste dysfunction, nausea, and vomiting (*p* > 0.05), for the 31 to 40 years group they were thirst, memory impairment and confusion (*p* > 0.05) (Appendix A). However, when analyzing the total number of symptoms, only the number of symptoms was significantly higher in the participants aged 11 to 20 years compared to the group aged 20 to 30 years (*p* > 0.05) (Figure 3). 

#### 3.2.4. Long-COVID Symptoms by Vaccination Status

Those patients with complete vaccine schemes and a single booster comprised the majority of Long-COVID reports 70.6% (*n* = 5472), however, the highest proportion of symptoms was observed in the group that received less than 2 doses (no vaccine and 1 dose) compared to the rest of the groups (*p* > 0.05) (Table 2 and Figure 4a).

On the other hand, the influence of the time of infection in relation to the time of receiving the vaccine showed that participants who were infected before receiving the vaccine had overall more symptoms 53.7% (*n* = 4164), this increase was represented by hair loss, loss of smell, loss of taste, difficulty concentrating and alterations in taste (*p* > 0.05), while among participants who were infected after being vaccinated, persistent cough was higher (*p* > 0.05) (Table 2).

Regarding to the participants who were infected after the vaccine (*n* = 539), the proportion of symptoms was lower in the group that reported having received 2 doses and 2 boosters compared to the groups with fewer vaccinations received (Figure 4b).

After performing a principal component analysis, we can conclude that LC symptoms behave the same in all people who have two or more doses of vaccines (Figure 5a) and there is no clear pattern on the difference in how symptoms are related to a given vaccination schedule. Furthermore, although there is no clear difference in how the vaccination schedule is related to LC symptoms, when grouping similar symptoms together, it was evident that LC symptoms affected participants regardless of the vaccination schedule they have (Figure 5b).

### 3.3. High Altitude of Residence

Overall, the population residing at high altitude reported a slightly higher proportion 14.3% (5503 symptoms in 789 participants) of Long-COVID symptoms when compared to the low-landers 13.7% (2228 symptoms in 306 participants) (*p* < 0.001) (Figure 6a).

According to sex distribution, in low altitudes residents a higher proportion of Long-COVID symptoms was found in men (*p* = 0.003), opposite tendency to that found at high altitudes (Figure 6b).

On various Long-COVID symptoms such as mood changes, tachycardia, decreased libido/sexual desire, and insomnia, a higher frequency was found in high altitude participants (*p* < 0.05) (Table 3). We also found a significantly earlier onset of symptoms (*p* = 0.004) in high altitude participants. Additionally, when comparing the proportion of patients having sequalae that last longer than a year, high altitude dwellers have a higher probability (*p* = 0.007) to report symptoms after 12 months (Table 3).

### 3.4. Symptoms Onset, Length, and Duration

The duration of Long-COVID symptoms shows a similar trend between men and women. (Appendix A). The most self-reported Long-COVID symptoms and those that last the longest are fatigue, loss of taste, loss of smell, and persistent cough (Appendix A). In terms of duration, 24.5% (*n* = 1899) of self-reported symptoms have been experienced at least for 12 months daily in 25.9% of patients (*n* = 2003), and more than three days per week in 29.5% of respondents (*n* = 2284) (Appendix A).

### 3.5. Risk of Long-COVID Symptoms

Based on the data from all respondents (*n* = 2103), the risk of developing Long-COVID symptoms in Ecuadorians was higher among women OR = 1.31 (95%CI: 1.097–1.558) compared to men, as well as in patients who reported having comorbidities OR = 1.95 (95%CI: 1.541–2.453) compared to those who did not. A significant higher risk of having post-acute symptoms was also observed among those with severe infection history OR = 7.79 (95%CI: 2.344–25.878) when compared to those who had a milder infection (Table 4).

### 3.6. Risk by Number of COVID-19 Infections

Among participants with Long-COVID symptoms (*n* = 1100), the effect of the number of previous infections as a risk factor for prolonged duration of Long-COVID symptoms was evident; participants who reported a history of more than 1 COVID-19 infection had a higher risk (RR = 1.296; 95%CI:1.076–1.560) of having sequelae lasting longer than 6 months (Appendix A). Similarly, the group of participants who claimed to have a history of more than 1 infection had an increased risk (RR = 1.329; 95%CI:1.042–1.696) of having sequelae with duration greater than 12 months (Appendix A).

### 3.7. Health-Related Quality of Life of Participants with Post-Acute COVID

Health-related quality of life of participants with Long-COVID was measured using the EQ-5D-3L. Overall, 8.0% of participants reported problems with self-care, 13.0% problems with mobility, 31.2% problems with usual activities, 81.4% with pain/discomfort and 98.9% reported experiencing anxiety and depression. The average index value for health status was 0.84 (SD = 0.08) and it was negatively correlated with number of experienced symptoms (r = −0.14). The average VAS score was 75.27 (SD = 21.34) and it was also negatively correlated with the number of symptoms (r = −0.21).

## 4. Discussion

To the best of our knowledge, this is the first study in South America to examine the Long-COVID symptoms of SARS-CoV-2 virus infection in a population with and without a history of hospitalization due to infection. This analysis revealed that women, young adults, overweight patients, patients without comorbidities, participants who were infected prior to vaccination, and participants who have received incomplete vaccine schemes developed more symptoms of long COVID.

The participants of this research mostly claimed to have a confirmed diagnosis through laboratory tests or medical examination (83.9% and 3.3%), however, the answers from those who claimed to have been infected by continuity (contagion by coexistence with an infected person with a confirmed diagnosis) were included as valid. Despite representing only, a small portion of Ecuador’s population (*n* = 2103), this study showed that just over half (52.3%) of participants with a history of COVID-19 infection developed at least 1 long-term sequelae of the infection. This corroborates the results found by Walsh-Messinger. et al. where a 51% prevalence of “post COVID syndrome” in self-reported university students who experienced mild to moderate infection (not hospitalized) was reported [25], as well as a self-reported prevalence of 60% among Spanish participants with a history of hospitalization [26]. On the other hand, a self-reported investigation of 310 individuals from several countries (Norway, United Kingdom, United States and Australia) showed a lower prevalence (28.7%) of long-COVID sequalae when compared to our results [27], nevertheless, in a systematic review a prevalence of 43% was estimated at a global level [28].

We were able to show that the participants managed to describe the existence of 52 different symptoms in a total of 7746 occasions. These consequences affected several organs and systems, causing health problems not only in the physical aspect but also in the neurocognitive and emotional aspect. The most frequently reported Long-COVID symptoms were fatigue or tiredness (8.4%), followed by hair loss (5.0%) and difficulty concentrating (5.0%). These findings are similar to those found in meta-analyses and large population studies that place fatigue and hair loss among the most prevalent post-COVID symptoms [28,29,30]. Similarly, in studies that based their analyses on self-reported information on neurological sequelae, Bungenberg et al. found that the most frequent symptoms were difficulties in attention and concentration, and fatigue among hospitalized and non-hospitalized participants [31], while the results in patients with history of hospitalization showed that the most frequent symptoms were memory deficit and attention deficit [32].

Regarding sex, it was evidenced that women presented more Long-COVID symptoms (feeling of anxiety, brittle hair, difficulty concentrating, headache, insomnia, nausea, neuritis, hair loss, memory defects, muscle loss, dry skin, easy crying, and brittle nails) when compared to men, findings similar to those presented by Sudre et al. and Whitaker et al. in self-report studies in the population of the United Kingdom [33,34], Mexico [35] and Spain [26]. While these differences may be because women are more likely to participate in this type of investigations and are more aware of their health, only one study has shown that men are more likely to develop long-COVID than women [27,36]. 

With respect to the age of the participants, we observed that the Long-COVID symptoms are more likely reported by young adults between 20 and 40 years old. A total of 21 symptoms including taste dysfunction, confusion, tingling in the extremities, nausea, memory defects, among others showed to be significantly more frequent in this younger group compared to the groups of other ages. These results corroborate that of a similar study in the UK population with no history of hospitalization and in which the risk of reporting symptoms of Long-COVID decreases with age [30]. On the other hand, of those who reported Long-COVID (*n* = 1100), only *n* = 61 (5.6%) participants younger than 20 years reported Long-COVID symptoms, close to that reported in a prospective cohort study in Moscow, Russia, where a much smaller portion (12.6%) of Long-COVID sufferers were children; the same difference in distribution was observed in the Italian population [37,38]. In the same context, among our responders under 20 years of age, the most reported symptoms were fatigue, loss of smell, difficulty concentrating, and anxiety, while Pazukhina et al. found that at 6- and 12 months post-infection the most frequent symptoms in Russian children were fatigue, dermatological and neurocognitive symptoms [37].

The effect of comorbidities on the self-reported results reveals unexpected behavior within our sample of respondents. In particular, a higher frequency of Long-COVID symptoms was observed in participants who claimed to have no history of comorbidities than those with comorbidities. However, in the risk analysis comparing the group of patients without symptoms of Long-COVID, it was found that having comorbidities is a risk factor for the development of post-acute symptoms (OR = 1.95; 1.541–2.453). These discrepancies are possibly caused by most of the responders not being diagnosed with comorbidities. The effects of concomitant diseases have been seen in the studies of Subramanian et al., and Fernández-de-las-Peñas et al., in populations of United Kingdom and Spain, respectively [30,39]. Similarly, in relation to the body mass index of the participants, those who reported having a normal BMI (18.5–24.9) had a higher number of Long-COVID symptoms compared to the rest of the groups (*p* > 0.05), however, in a separate analysis it was evidenced that peripheral symptoms such as decreased strength, burning sensation in the body, tingling of limbs and reflux were significantly more frequent in participants in the overweight group compared to the others. Research in participants with and without a history of hospitalization in England, as well as in non-hospitalized groups in United Kingdom, has found overweight and obesity as risk factors for the development of Long-COVID symptoms (HRa = 1.07; 1.04–1.10) (HRa = 1.10; 1.07–1.14), respectively [30,34].

In relation to ethnicity, the mestizo (mixed) ethnic group showed significantly more Long-COVID symptoms than the rest of the ethnic groups. However, again, we believe that this result may reflect the fact that the largest number of responders were from this group, consistent with the fact that the mestizo ethnic group is the most prevalent in Ecuador (71.9%) [40]. In any case, we consider this interesting finding to be supported by the results presented by Subramanian et al. based on data from a large population study in the United Kingdom, in which being of mestizo ethnicity was associated with an increase in risk (HRa = 1.14; 1.07–1.22) of developing symptoms of Long-COVID taking as a reference group the white ethnic group [30], however, most of the reports published so far base their results on the analysis of populations with groups that have almost no mestizo individuals, unlike the Ecuadorian population that was investigated in this report.

Regarding habits, the risk analysis stated that alcohol consumption and smoking do not represent a risk condition for the development of Long-COVID symptoms after the infection by the SARS-CoV-2 virus. However, several reports from online surveys have shown that the habit or history of smoking increases the risk of Long-COVID symptoms especially tachycardia and/or hypertension in the case of population of France, as well as in inhabitants of the United Kingdom (HRa = 1.12; 1.08–1.15), (HRa = 1.08; 1.05–1.11) [30,34,41].

The effect of the severity of the infection (antecedent) characterized by the treatment that the participants claimed to receive showed that in contrast to responders who did not develop sequelae, having presented severe infection pictures (hospitalization for more than 7 days and/or hospitalization in the intensive care unit) was a great risk condition (OR = 7.79; 2.34–25.88) for the development of Long-COVID sequelae compared with mild infection (outpatient treatment and non-treatment). Similar effects have been evidenced in various studies characterized by the use of mechanical ventilation, and having been hospitalized [28,32,34,35]. 

The evaluation of the effect of vaccines on Long-COVID symptoms showed that respondents with two of the recommended boosters are less likely to develop long term symptoms when compared to those with incomplete vaccination status. As users of two doses plus two boosters showed a lower number of Long-COVID symptoms. Despite the protective effects that vaccines have shown for the development of Long-COVID in other studies [42,43], our category analysis showed limited or no effect of vaccines in relation to the occurrence of reported LC symptoms, studies with more robust methodology would be essential to explain the role of vaccines in the development of LC.

In the same context, when assessing the role of the time individuals acquire infection in relation to the timing of vaccination, we found that the frequency of Long-COVID symptom presentation was significantly higher in participants who acquired the infection before being vaccinated. The symptoms in non-vaccinated patients were frequently difficulty concentrating, decreased visual acuity, hair loss, loss of taste and loss of smell. Alternatively, for participants who acquired the infection after being vaccinated with two doses plus two boosters of the vaccine, there was found to be a significant decrease in the number of Long-COVID symptoms. A partially similar effect was described by Ayoubkhani et al. in the United Kingdom, who demonstrated a decrease in Long-COVID symptoms in participants who were infected before vaccination [43].

We observed that the duration of symptoms reported was variable, but in many cases greater than 12 months (24.5%). The analysis of the time of onset and the frequency of presentation of symptoms incorporated extra aspects relative to those incorporated in the definition of a “long COVID case” of daily presentation proposed by the WHO [8], thus, we found that the highest number of Long-COVID symptoms had onset with the virus infection (42.1%), although not large in number, out study collected information regarding symptoms that began up to 9 weeks after suffering the infection. The present study also discovered that the majority of participants said that these persistent symptoms occur on more than 3 days per week (29.5%) and often daily (25.9%). This latter trait was also observed in participants from the United Kingdom and the United States who reported that the greatest number of symptoms occurs daily 72.2% and on more than three occasions per week 18.8% [44].

The products of the behavior analysis of the symptoms separately reveals that the symptom onset time can differ by symptom. Specifically, the onset of symptoms together with the infection exposed a significantly higher frequency in the diffusion of taste, diarrhea, fatigue, loss of appetite, loss of taste, loss of smell and persistent cough. These results are unlike the findings of Ziauddeen et al., where data from 2550 non-hospitalized participants from the United Kingdom and the United States found that the most frequent long-lasting initial symptoms were exhaustion, chest pressure difficulty and headache [44]. Regarding the evolution of the symptoms, our participants presented widely distributed evolution traits. The least reported pattern of symptom evolution was the worsening of the symptoms from their presentation (5.6%), while the intermittent pattern was the most common (32.1%). In such cases the presentation was characterized by symptoms significantly greater than the rest of the patterns such as tachycardia, decreased visual acuity, loss of muscle strength, decreased sex drive, chest pain, dry skin, easy crying; all behavior similar to that described by Ziauddeen et al. where the majority of participants (57.7%) described fluctuating symptoms [44]. In relation to the duration of symptoms, the most frequently reported symptoms of a prolonged duration were chest pain and persistent cough, which lasted mostly between 4 and 8 weeks. On the other hand, there are many symptoms of a psychic origin (confusion, difficulty concentrating, speech difficulty, memory deficit), of neuronal origin (loss and dysfunction of taste, loss of smell), and considered peripheral (intolerance to cold, tingling of the limbs, hair loss) that have been reported to persist with a duration greater than 12 months from the infection. This differs from the self-report study of Sudre et al. that showed that loss of smell, shortness of breath and fatigue were the symptoms of greater duration, however, that the duration of these was found to be only between 40 and 70 days approximately [33].

Symptom characteristics, such as time of onset, duration, and frequency of presentation, were little altered by many variables studied (sex, comorbidities, vaccination, ethnicity, age). However, a unique result from the present study shows that the altitude analysis reveals that the post-acute symptoms of the participants residing in areas located above 2500 m tend to occur with lower frequencies (once a week). Nevertheless, while the frequency of the symptom presentation is lower, the duration of symptoms in this group is apparently longer compared to that observed in residents below 2500 m. Although there are distinct results in patients by altitude, the risk analysis did not expose altitude as a risk condition for the development of post-acute symptoms of COVID-19. Despite certain differences described during the acute course of infection in high-altitude and low-altitude populations such as incidence and mortality [22,23], this is the first time that this variable is included in the study of Long-COVID symptoms of COVID-19.

Likewise, according to the development of Long-COVID symptoms, people at high altitudes (>2500 m) showed more symptoms such as mood swings, tachycardia, decreased libido, insomnia, and palpitations compared to people at low altitudes, although the influence of hypobaric hypoxia on Long-COVID has not yet been studied, we could posit a confounding effect of environmental contamination on the development of sequelae. It has been shown that terrains with higher levels of contamination promote higher rates of severe COVID-19 infection, and, according to 2021 data the city with the highest contamination levels in Ecuador was Quito, a city located at high altitude (2850 m) [45]. This opens the possibility that the significant differences found in the development of Long-COVID symptoms in high-altitude residents may be altered by air pollution levels, as well as by the level of air pollution in the city. 

Along the same line, during the initial stages of the pandemic, it was erroneously proposed that acute mountain sickness (AMS), a condition that brings together a wide range of symptoms including headache, loss of appetite, nausea, vomiting, fatigue, sleep disturbances and COVID-19 infection may share some pathophysiological mechanisms that are involved in the development of these conditions [46]. In this context, much of the symptoms of acute mountain sickness are attributed to the acute exposure to high-altitude that is often quick and sustained among not adapted or no acclimatized subjects [47]. Beside this, AMS last as long as the exposure is sustained, being easily treated when returning to lower altitudes [48]. In our participants, some Long-COVID symptoms occurred for long periods of time, and the proposed mechanisms is unknown, it might be linked to linked to psychological and emotional characteristics that make high altitude dwellers more prone to complain about their health than low altitude dwellers [49].

More research is needed to estimate the prevalence of Long-COVID symptoms associated with the history of infection by the SARS-CoV-2 virus among inhabitants of South America. In current studies, the role of ethnic differences during the sequelae of the disease could be underestimated because of the scarcity of information from populations with a large degree of miscegenation. Future studies could also examine the wide range of sociodemographic factors that have been associated with the development of these symptoms, of which the South American region has marked differences in its inhabitants.

Our study has several limitations which are inherent to its self-reported cross-sectional design. Since the questionnaire was distributed through social networks, information pertaining to the population that does not have the resources to access an online questionnaire was left out of the study. Likewise, most older adults handle electronic devices and the Internet with difficulty. This may cause a selection bias. However, we believe that this bias was minimized by using a substantial sample size taken from the data of the Ecuadorian population that was infected by the SARS-CoV-2 virus. Another limitation of this type of study is the absence of accurate and verified past medical history, as in the case of severity of infection. Nevertheless, we attempted to assess these variables by using more objective questions such as general characteristics of hospitalization that are more likely to be accurately captured by participants. In addition, this research was exposed to social desirability bias characterized by a possible inclination of respondents to ensure that they suffer from Long-COVID symptoms, however, we tried to avoid the effects of this bias by providing in several questions of the questionnaire the option of not having suffered from COVID-19 infection, as well as the option of not suffering from Long-COVID, we also purified the answers that showed contradictions in relation to these questions and being a voluntary access questionnaire we believe that the results found can be very close to reality. On the other hand, unlike other studies that similarly sought to evaluate Long-COVID symptoms of COVID-19 from diagnoses confirmed by laboratory studies, in this research we accepted cases of patients who claimed to have contracted the disease by continuity. This is important in a country like Ecuador where the social reality allows for a limited capacity of access to confirmatory laboratory tests for the population. Social desirability bias is another potential limitation that could have affected the responses due to the self-report nature of the questionnaire.

## 5. Conclusions

This is the first study on the long-term sequelae of COVID-19 in Ecuador and, to our knowledge, the first of its kind in South America. The results of this study on self-reported symptoms of Long-COVID show that in Ecuador, at least half of those previously infected with SARS-CoV-2 have at least one persistent symptom after overcoming the disease. From the self-reported list of Long-COVID symptoms, fatigue, anosmia, ageusia, hair loss and difficulty concentrating are among the most common ongoing conditions, lasting in many cases more than a year after COVID-19. We found that young adults, especially those with past medical history of severer COVID-19 and those with more than one infection are more likely to report having more sequalae that persist longer than those with no previous infection. High altitude dwellers have a slightly greater probability to report mood changes, tachycardia, decreased libido, insomnia, and palpitations, moreover, in low altitudes residents, men have a higher Long-COVID symptoms.

## Figures and Tables

**Figure 1 ijerph-19-14673-f001:**
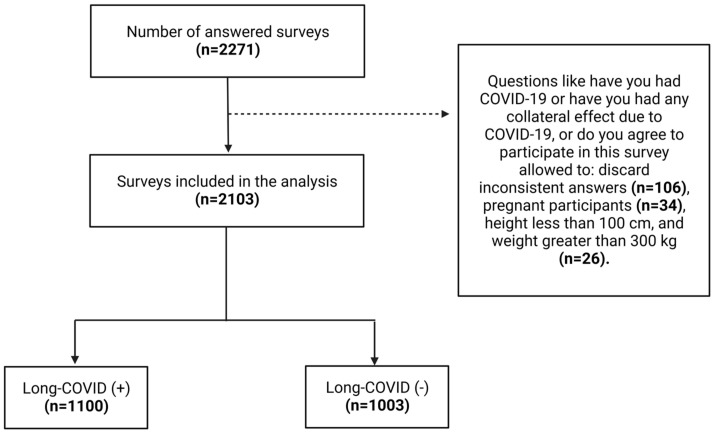
Breakdown of included and excluded surveys.

**Figure 2 ijerph-19-14673-f002:**
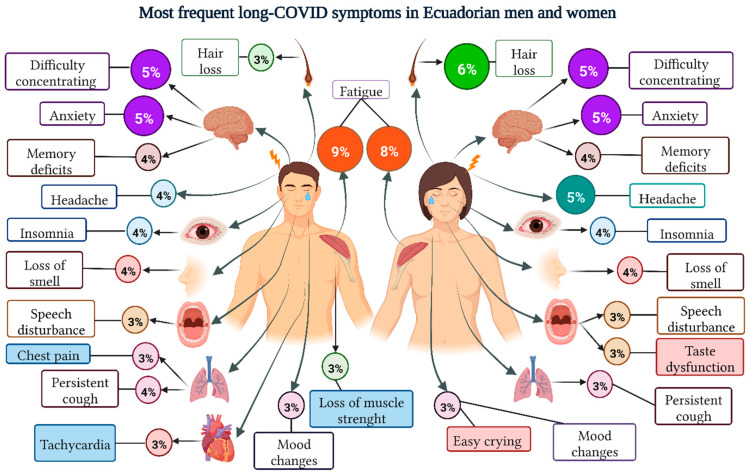
Prevalence of most frequent Long-COVID symptoms in Ecuadorian participants according to sex.

**Figure 3 ijerph-19-14673-f003:**
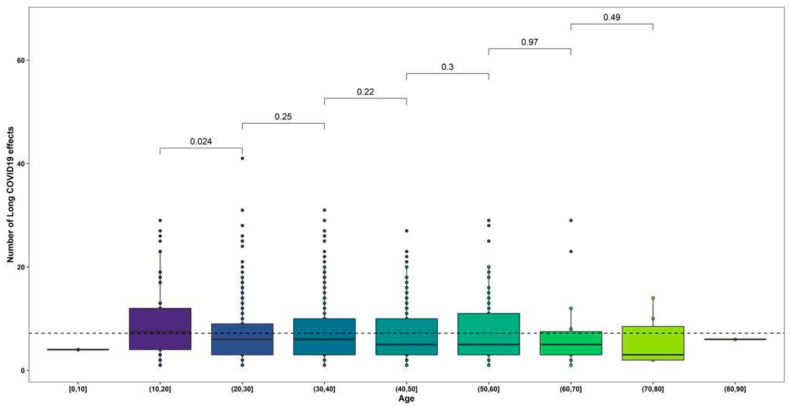
*t*-test between number of Long-COVID symptoms and age groups of participants.

**Figure 4 ijerph-19-14673-f004:**
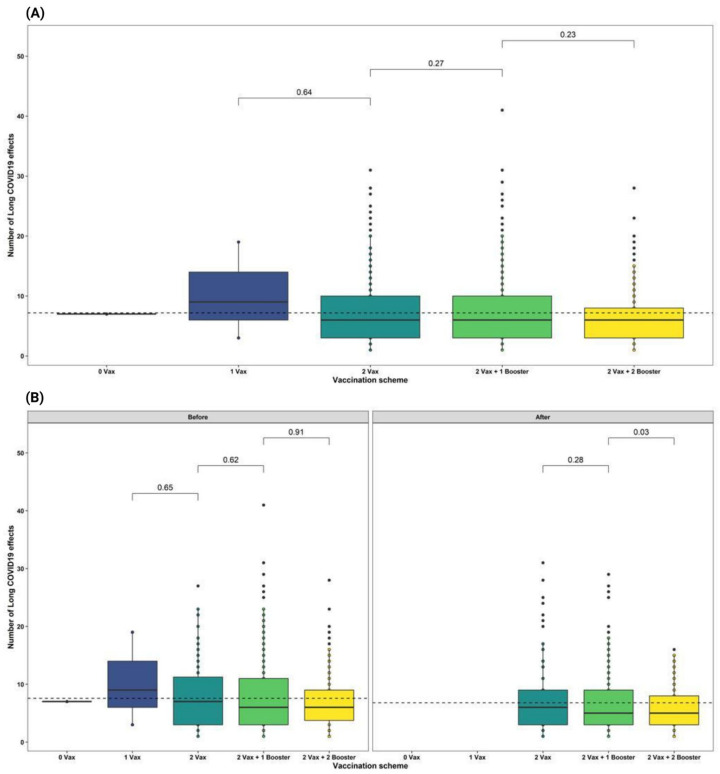
Effect of vaccination on Long-COVID symptoms in Ecuadorian participants. (**A**). *t*-test between number of Long-COVID symptoms and vaccine received doses. (**B**). *t*-test between number of symptoms and vaccine doses received according to the time of infection (left: infection before vaccine; right: infection after vaccine).

**Figure 5 ijerph-19-14673-f005:**
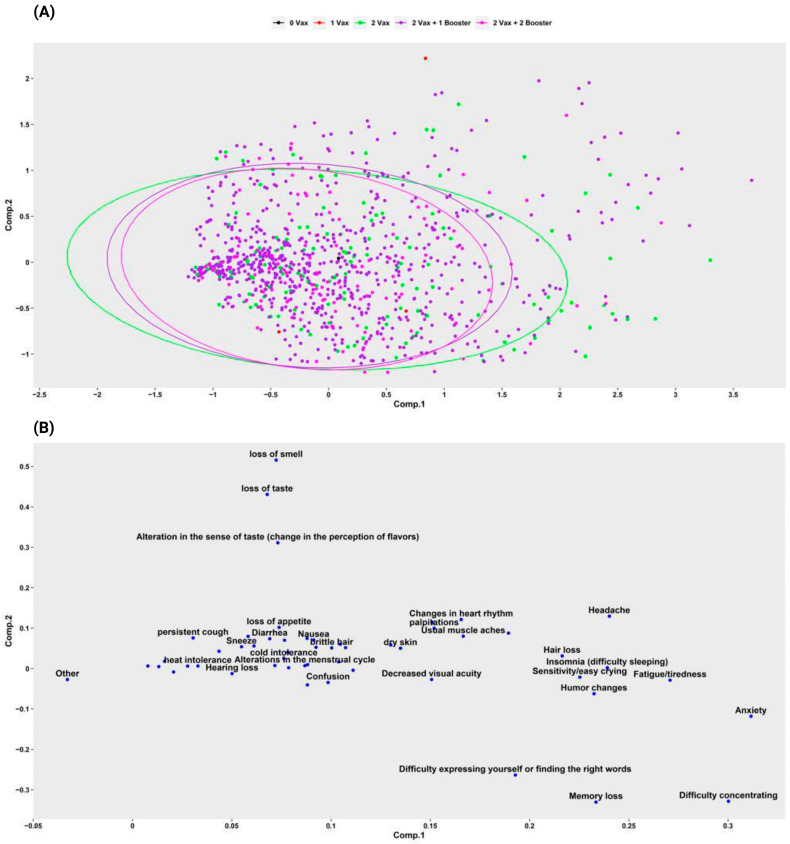
Multicomponent analysis of Long-COVID symptoms and vaccines received. (**A**). Side effects (Long-COVID symptoms) behave symmetrically in participants with at least two doses of vaccines. (**B**). The clustered symptoms affected participants regardless of the vaccination schedule.

**Figure 6 ijerph-19-14673-f006:**
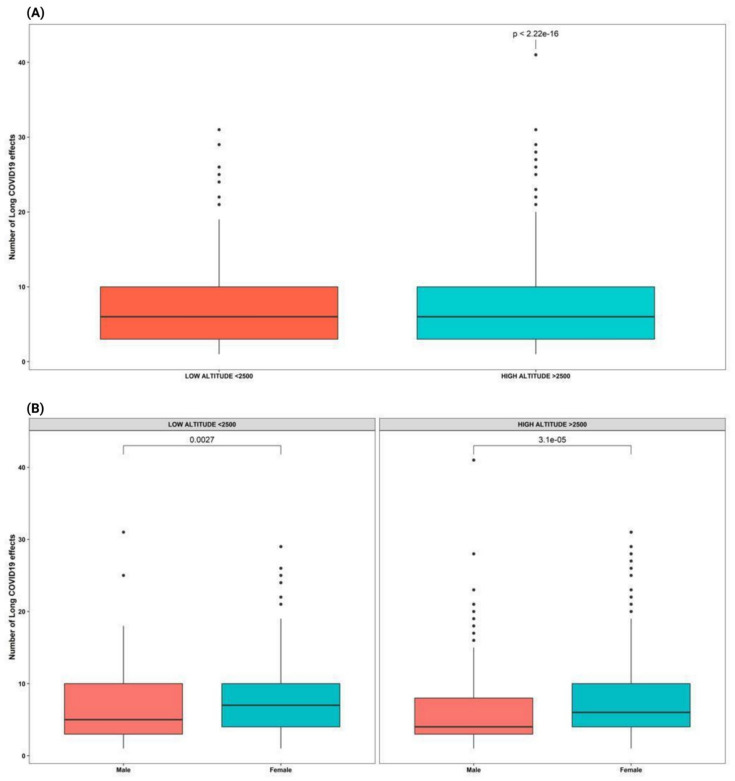
Differences in Long-COVID symptoms according to altitude. (**A**) *t*-test between the number of long-COVID symptoms and altitude. (**B**) *t*-test between the number of long-COVID symptoms and altitude according to sex.

**Table 1 ijerph-19-14673-t001:** Overall characteristics of participants with COVID-19 history. Bold *p*-values are statistically significant.

Characteristics		Long COVID *n* (%)	Not Long COVID *n* (%)	Total *n*	*p*-Value
Respondents		1100 (52.3)	1003 (47.7)	2103	
**Demographics**					
Sex	Male	396 (48.2)	425 (51.8)	821	**0.003**
	Female	704 (54.9)	578 (45.1)	1282	
Age (years)	0 to 10	1 (33.3)	2 (66.7)	3	**<0.001**
	10 to 20	60 (33.5)	119 (66.5)	179	
	21 to 30	382 (50.1)	381 (49.9)	763	
	31 to 40	372 (56.8)	283 (43.2)	655	
	41 to 50	191 (55.7)	152 (44.3)	343	
	51 to 60	72 (57.1)	54 (42.9)	126	
	61 to 70	15 (60.0)	10 (40.0)	25	
	71 to 80	6 (75.0)	2 (25.0)	8	
	81 to 90	1 (100.0)		1	
Residence altitude	Out of the country	5 (55.6)	4 (44.4)	9	0.368
	>2500 m	789 (53.3)	692 (46.7)	1481	
	<2500 m	306 (49.9)	307 (50.1)	613	
Comorbidities	No	853 (49.4)	873 (50.6)	1726	**<0.001**
	Yes	247 (65.5)	130 (34.5)	377	
Smoke	No	993 (52.4)	901 (47.6)	1894	0.79
	Yes	107 (51.2)	102 (48.8)	209	
Alcohol	No	298 (53.0)	264 (47.0)	562	0.726
	Yes	802 (52.0)	739 (48.0)	1541	
Severity of infection	Mild	1056 (51.7)	987 (48.3)	2043	**<0.001**
	Moderate	19 (59.4)	13 (40.6)	32	
	Severe	25 (89.3)	3 (10.7)	28	

**Table 2 ijerph-19-14673-t002:** Distribution of self-reported Long-COVID symptoms according to vaccination. Bold *p*-values are statistically significant.

Symptom		Vaccine *n* (%)					Time of Infection *n* (%)			
	Total *n*	No Vax.	1 Dose	2 Doses	2 Doses + 1 Boosters	2 Doses + 2 Boosters	*p*-Value	Before Vaccine	After Vaccine	*p*-Value
Alopecia	32	--	--	7 (21.9)	23 (71.9)	2 (6.2)	0.383	16 (50.0)	16 (50.0)	0.999
Taste dysfunction	199	--	1 (0.5)	32 (16.1)	138 (69.3)	28 (14.1)	0.841	121 (60.8)	78 (39.2)	**0.002**
Alterations in glucose	16	--	--	5 (31.3)	10 (62.5)	1 (6.2)	0.202	9 (56.2)	7 (43.8)	0.864
Menstrual cycle alterations	118	--	--	16 (13.6)	81 (68.6)	21 (17.8)	0.231	56 (47.5)	62 (52.5)	0.473
Hallucinations	20	1 (5.0)	--	6 (30.0)	13 (65.0)	--	**<0.001**	13 (65.0)	7 (35.0)	0.299
Anxiety	379	1 (0.3)	1 (0.3)	68 (17.9)	271 (71.5)	38 (10.0)	0.114	205 (54.1)	174 (45.9)	0.154
Brittle hair	111	--	--	16 (14.4)	81 (73.0)	14 (12.6)	0.901	65 (58.6)	46 (41.4)	0.114
Mood changes	234	--	--	40 (17.1)	173 (73.9)	21 (9.0)	0.119	123 (52.6)	111 (47.4)	0.641
Tachycardia	184	1 (0.5)	1 (0.5)	32 (17.4)	131 (71.3)	19 (10.3)	0.13	99 (53.8)	85 (46.2)	0.451
Blood pressure changes	74	--	1 (1.3)	13 (17.6)	51 (68.9)	9 (12.2)	0.303	42 (56.8)	32 (43.2)	0.365
Confusion	61	1 (1.6)	--	18 (29.5)	39 (64.0)	3 (4.9)	**<0.001**	39 (63.9)	22 (36.1)	0.051
Diarrhea	56	--	--	10 (17.9)	36 (64.2)	10 (17.9)	0.436	32 (57.1)	24 (42.9)	0.419
Difficulty concentrating	387	--	2 (0.5)	59 (15.2)	270 (69.8)	56 (14.5)	0.449	214 (55.3)	173 (44.7)	**0.042**
Speech disturbance	241	--	1 (0.4)	45 (18.7)	159 (66.0)	36 (14.9)	0.26	126 (52.3)	115 (47.7)	0.705
Decreased visual acuity	147	--	1 (0.7)	29 (19.7)	100 (68.0)	17 (11.6)	0.361	92 (62.6)	55 (37.4)	**0.003**
Loss of muscle strength	170	--	--	32 (18.8)	128 (75.3)	10 (5.9)	**0.009**	91 (53.5)	79 (46.5)	0.526
Decreased libido/sexual desire	115	--	--	18 (15.7)	81 (70.4)	16 (13.9)	0.949	60 (52.2)	55 (47.8)	0.866
Abdominal pain	70	--	--	11 (15.7)	47 (67.2)	12 (17.1)	0.551	32 (45.7)	38 (54.3)	0.429
Headache	354	1 (0.3)	1 (0.3)	56 (15.8)	246 (69.5)	50 (14.1)	0.578	184 (52.0)	170 (48.0)	0.702
Chest pain	187	--	1 (0.5)	31 (16.6)	136 (72.7)	19 (10.2)	0.555	92 (49.2)	95 (50.8)	0.644
Burning sensation in any part of the body	75	--	--	15 (20.0)	57 (76.0)	3 (4.0)	**0.046**	40 (53.3)	35 (46.7)	0.764
Unusual muscle aches	109	--	1 (0.9)	18 (16.5)	77 (70.7)	13 (11.9)	0.576	56 (51.4)	53 (48.6)	0.999
Usual muscle aches	164	--	1 (0.6)	33 (20.1)	112 (68.3)	18 (11.0)	0.269	88 (53.7)	76 (46.3)	0.513
Shaking chills	45	--	1 (2.2)	8 (17.8)	33 (73.3)	3 (6.7)	**0.043**	22 (48.9)	23 (51.1)	0.891
Sneezing	92	--	1 (1.0)	11 (12.0)	71 (77.2)	9 (9.8)	0.199	41 (44.6)	51 (55.4)	0.237
Fatigue or tiredness	652	1 (0.2)	2 (0.3)	101 (15.5)	460 (70.5)	88 (13.5)	0.863	331 (50.8)	321 (49.2)	0.9
Tingling in extremities	137	--	--	25 (18.3)	101 (73.7)	11 (8.0)	0.159	76 (55.5)	61 (44.5)	0.303
Insomnia	342	--	--	54 (15.8)	252 (73.7)	36 (10.5)	0.262	175 (51.2)	167 (48.8)	0.991
Heat intolerance	20	--	--	6 (30.0)	12 (60.0)	2 (10.0)	0.211	10 (50.0)	10 (50.0)	0.999
Cold intolerance	92	--	--	16 (17.4)	65 (70.6)	11 (12.0)	0.883	54 (58.7)	38 (41.3)	0.151
Nausea	68	--	1 (1.5)	13 (19.1)	47 (69.1)	7 (10.3)	0.19	31 (45.6)	37 (54.4)	0.425
Neuritis	96	--	--	20 (20.8)	65 (67.7)	11 (11.5)	0.355	48 (50.0)	48 (50.0)	0.921
Palpitations	182	1 (0.6)	1 (0.5)	30 (16.5)	125 (68.7)	25 (13.7)	0.197	96 (52.8)	86 (47.2)	0.663
Facial paralysis	7	--	--	1 (14.3)	5 (71.4)	1 (14.3)	0.99	5 (71.4)	2 (28.6)	0.48
Loss of appetite	55	--	1 (1.8)	9 (16.4)	39 (70.9)	6 (10.9)	0.153	23 (41.8)	32 (58.2)	0.207
Hearing loss	49	--	--	7 (14.3)	36 (73.5)	6 (12.2)	0.937	33 (67.3)	16 (32.7)	**0.028**
Hair loss	391	--	2 (0.5)	62 (15.9)	274 (70.0)	53 (13.6)	0.672	227 (58.1)	164 (41.9)	**<0.001**
Loss of taste	172	--	1 (0.6)	27 (15.7)	125 (72.7)	19 (11.0)	0.719	114 (66.3)	58 (33.7)	**<0.001**
Memory deficits	287	--	2 (0.7)	54 (18.8)	196 (68.3)	35 (12.2)	0.14	155 (54.0)	132 (46.0)	0.264
Muscle loss	97	--	1 (1.0)	12 (12.4)	69 (71.1)	15 (15.5)	0.315	59 (60.8)	38 (39.2)	0.054
Loss of smell	263	--	2 (0.8)	37 (14.1)	190 (72.2)	34 (12.9)	0.296	170 (64.6)	93 (35.4)	**<0.001**
Body hair loss	14	--	--	2 (14.3)	12 (85.7)	--	0.978	7 (50.0)	7 (50.0)	0.999
Pruritus (itching)	114	--	--	22 (19.3)	76 (66.7)	16 (14.0)	0.475	66 (57.9)	48 (42.1)	0.145
Gastroesophageal reflux	87	--	--	16 (18.4)	62 (71.3)	9 (10.3)	0.636	51 (58.6)	36 (41.4)	0.17
Skin dryness	139	--	--	28 (20.1)	98 (70.5)	13 (9.4)	0.171	71 (51.1)	68 (48.9)	0.999
Thirst	77	--	1 (1.3)	11 (14.3)	61 (79.2)	4 (5.2)	**0.048**	30 (39.0)	47 (61.0)	**0.038**
Easy crying	202	--	--	40 (19.8)	138 (68.3)	24 (11.9)	0.217	114 (56.4)	88 (43.6)	0.102
Excessive sweating	90	--	1 (1.1)	16 (17.8)	59 (65.5)	14 (15.6)	0.292	51 (56.7)	39 (43.3)	0.311
Tremor of the extremities	90	--	--	20 (22.2)	59 (65.6)	11 (12.2)	0.216	48 (53.3)	42 (46.7)	0.724
Persistent cough	245	--	1 (0.4)	28 (11.4)	185 (75.5)	31 (12.7)	0.184	93 (38.0)	152 (62.0)	**<0.001**
Brittle nails	113	--	--	20 (17.7)	80 (70.8)	13 (11.5)	0.778	58 (51.3)	55 (48.7)	0.999
Vomit	25	--	1 (4.0)	4 (16.0)	17 (68.0)	3 (12.0)	**0.004**	10 (40.0)	15 (60.0)	0.362
Total	**7746**	**7**	**31**	**1310**	**5472**	**926**		**4164**	**3582**	

**Table 3 ijerph-19-14673-t003:** Long-COVID symptoms and their characteristics according to the altitude of the participants’ residence.

		Altitude n (%)				
		Out of the Country	<2500 m	>2500 m	*p*-Value	Total *n*
Symptoms	Taste dysfunction	--	46 (23.1)	153 (76.9)	0.111	199
	Menstrual cycle alterations	1 (0.9)	26 (22.0)	91 (77.1)	0.276	118
	Anxiety	1 (0.3)	114 (30.0)	264 (69.7)	0.392	379
	Brittle hair	--	33 (29.7)	78 (70.3)	0.741	111
	Mood changes	--	79 (33.8)	155 (66.2)	**0.031**	234
	Tachycardia	--	67 (36.4)	117 (63.6)	**0.006**	184
	Difficulty concentrating	1 (0.3)	101 (26.1)	285 (73.6)	0.485	387
	Speech disturbance	--	59 (24.5)	182 (75.5)	0.202	241
	Decreased visual acuity	1 (0.7)	44 (29.9)	102 (69.4)	0.743	147
	Loss of muscle strength	1 (0.6)	54 (31.8)	115 (67.6)	0.433	170
	Decreased libido/sexual desire	1 (0.9)	43 (37.4)	71 (61.7)	0.038	115
	Headache	--	98 (27.7)	256 (72.3)	0.951	354
	Chest pain	--	59 (31.6)	128 (68.4)	0.263	187
	Unusual muscle aches	--	27 (24.8)	82 (75.2)	0.505	109
	Usual muscle aches	--	46 (28.0)	118 (72.0)	0.999	164
	Fatigue or tiredness	3 (0.5)	188 (28.8)	461 (70.7)	0.66	652
	Tingling in extremities	--	43 (31.4)	94 (68.6)	0.39	137
	Insomnia	2 (0.6)	114 (33.3)	226 (66.1)	**0.02**	342
	Palpitations	--	66 (36.3)	116 (63.7)	**0.008**	182
	Hair loss	2 (0.5)	113 (28.9)	276 (70.6)	0.815	391
	Loss of taste	--	47 (27.3)	125 (72.7)	0.916	172
	Memory deficits	1 (0.4)	73 (25.4)	213 (74.2)	0.542	287
	Loss of smell	--	67 (25.5)	196 (74.5)	0.344	263
	Pruritus (itching)	--	35 (30.7)	79 (69.3)	0.56	114
	Skin dryness	--	32 (23.0)	107 (77.0)	0.199	139
	Easy crying	--	59 (29.2)	143 (70.8)	0.721	202
	Persistent cough	1 (0.4)	75 (30.6)	169 (69.0)	0.54	245
	Brittle nails	--	30 (26.6)	83 (73.4)	0.811	113
	**Total**					**6338**
Symptom characteristics						
Onset time	Initiated with infection	4 (0.9)	104 (23.2)	341 (76.0)	**0.004**	
	3 to 5 weeks after infection	1 (0.2)	126 (30.4)	287 (69.3)	0.243	
	5 to 7 weeks after infection	.	28 (27.2)	75 (72.8)	0.948	
	7 to 9 weeks after infection	.	15 (30.0)	35 (70.0)	0.865	
	After 9 weeks of infection	.	33 (39.3)	51 (60.7)	**0.022**	
Symptoms duration	Between 1 to 4 weeks	.	53 (23.7)	171 (76.3)	0.129	
	Between 4 to 8 weeks	.	47 (25.8)	135 (74.2)	0.543	
	Between 8 to 12 weeks	3 (2.6)	28 (24.4)	84 (73.0)	**0.001**	
	Between 3 to 6 months	.	60 (28.6)	150 (71.4)	0.889	
	Between 6 to 12 months	.	34 (26.0)	97 (74.1)	0.662	
	More than 12 months	2 (0.8)	84 (35.3)	152 (63.9)	**0.007**	
Frequency of presentation	Once a month	.	23 (23.7)	74 (76.3)	0.393	
	Once every two weeks	.	20 (23.8)	64 (76.2)	0.452	
	Once a week	.	75 (34.7)	141 (65.3)	**0.016**	
	Over 3 days a week	2 (0.7)	82 (27.6)	213 (71.7)	0.804	
	Daily	3 (0.9)	94 (28.4)	234 (70.7)	0.324	
	**Total**	**15**	**918**	**2367**		

M: meters above sea level.

**Table 4 ijerph-19-14673-t004:** Risk analysis for Long-COVID symptoms in Ecuadorian participants.

Characteristics		Long COVID *n* = 1100	Not long COVID *n* = 1003	OR (95% CI)
		(n)	(n)	
Sex	Men *(ref.)*	396	425	**1.31 (1.10–1.56)**
	Women	704	578	
Altitude	Low *(ref.)*	306	307	1.14 (0.95–1.38)
	High	789	692	
Comorbidities	No *(ref.)*	853	873	**1.95 (1.541–2.453)**
	Yes	247	130	
Smoke	No *(ref.)*	993	901	0.95 (0.72–1.27)
	Yes	107	102	
Alcohol	No *(ref.)*	298	264	0.96 (0.79–1.17)
	Yes	802	739	
Severity of infection	Mild *(ref.)*	1056	987	
	Moderate	19	13	1.37 (0.67–2.78)
	Severe	25	3	**7.79 (2.34–25.88)**

OR: Odds Ratio; CI: confidence interval.

## Data Availability

The dataset with the total responses can be obtained from the authors upon reasonable request. The questionnaire summary and its results are included in this article.

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
