# Peer review of "Long COVID at Different Altitudes: A Countrywide Epidemiological Analysis"

_ijerph, 2022, doi:10.3390/ijerph192214673_

Round 1

Reviewer 1 Report

The article by Izquierdo-Condoy et al. is the result of a lot of work. Based on an electronic questionnaire they evaluated post covid conditions in 1003 patients without Long Covid (LC) and 1100 patient who claimed to have at least one symptom defining Long Covid.  Their main and original conclusions are that “High altitude resident were more likely to report persisting symptoms versus those living at lower altitude and that the highest proportion of persisting symptoms was observed among those who received an incomplete vaccine scheme.”

These conclusions are not supported by the facts. There is a confusion all along the manuscript between the number of patients with LC /No LC and the number of symptoms of LC which makes difficult interpretation of data and support overinterpretation made by the authors  

When the variable studied is patient as reported in table 1, no difference is noted between patient living at high altitude versus low altitude. This is obviously confirmed in the risk analysis in table 5 where altitude is not a risk for LC in these patients.

In the symptomatologic analysis it is reported that among the 1100 patients that report LC “at least” 7746 events were collected and grouped into 52 symptoms. Among these symptoms’ mood changes, tachycardia, decrease libido, insomnia palpitation is reported more frequently in patient living at high altitude. This analysis should first be compared to NO long covid cohort as these features may be due to altitude alone. This univariate analysis should be also confirmed by multiple regression analysis as some of these factors may be linked such as mood, insomnia ... Moreover, Red Maca is a well know plant medication in South America reported as efficient on mood, decrease libido … (https://doi.org/10.3390/ph9030049 ) and was proposed as a treatment for COVID-19 and such medication might be a confounding factor.   Any way, if these symptoms are confirmed to be different between patient living in high and low altitude the conclusion proposed by the authors is not supported by the data and should be changed.

On vaccine it is more confusing. If the question is:  does vaccination protect from Long covid? the analysis should compare cohorts (LC and No LC) for the different vaccination scheme and period.

Here the question seems; is there any relationship between symptoms in long covid patient and vaccination scheme? In table 3 is reported the distribution of symptoms according to vaccination scheme in long covid cohort. The lecture of table 3 is ; hallucinations, confusion, loss of muscle strength, burning sensation in any part of the body, shaking chills, thirst and vomit are more frequently reported in LC patient receiving a complete vaccine scheme and a booster. This is interesting but this does not mean that vaccine is protective of LC. What is happening what does this mean? To solve the question a Multi Component Analysis should clarified the possible association between symptoms and vaccine scheme.

Terminology

Please homogenize the terms “Long covid” “Post-acute symptoms” and “Post COVID”… all along the manuscript

Several other terms have been used to describe prolonged symptoms following COVID-19 illness, such as "long COVID," "postacute sequelae of SARS-CoV-2 infection (PASC)," "postacute COVID-19," "chronic COVID-19," and "post-COVID syndrome" [9-13]. Despite the creation of case definitions, there are no widely accepted clinical diagnostic criteria for "long COVID" [14]. However, as of October 1, 2021, there is a new International Classification of Diseases, Tenth Revision, Clinical Modification (ICD-10) for unspecified post-COVID conditions, which is U09.9, which was approved by the CDC.

Specific comments

Abstract: Please rewrite appropriately

Page 2 line 63:  sinus?

Page 6 Table 2: there are to many data in the manuscript, and I suggest to put the table 2 in supplementary data file

Page 9 line 196 you cannot “identified at least …. “some events. Please precise how many events you identified.

Page 13 the end of table 3 does not fit the upper part and it is difficult to get the total of each column.  

In figure 3;4 and 5 it is not clear for me if axis of the orders are distribution of events or symptoms. Please clarify. The lecture is that the number or reported symptoms in patient with LC is lower when vaccinated for at least two doses.

However, participant receiving the vaccine after their COVID had more symptoms (53.7%). This suggests a comment. Does this is a social desirability bias (post covid vaccine and deleterious effect) or this is realty “true” and need complementary investigation.

Conclusion

This is an interesting work that merit to be published. However, there is a serious need to clarify the research questions.

Analysis of the number and distribution of symptoms is interesting, but very weak in terms of reliability and interpretation, especially in the situation of LC which is a disease mostly with subjective claims, many bias (including social desirability, insurance needs) ….. The strength of the paper is the good and precise description of symptoms on a large cohort of patient, which is in fact not so frequently reported allowing to complete the current knowledge on this difficult public heath problem.

I would add the role of vaccination scheme and the time before and after natural infection on the Post covid condition (at least one claim/no claim at all) in the 2103 patients. As only 3 patients in the LC cohort have received less than two doses (99.9% having received at least 2 doses) and hypothesizing that it is approximately the same in Non LC cohort, it seems likely that no difference will appear, but need to be investigated as the repartition of the vaccine scheme before/after may differ in the two group.

Author Response

To: Liam Liu,

Invited Editor IJERPH

Manuscript ID: ijerph-1896361

Title: Long COVID at low and high altitude: A countrywide epidemiological

analysis of self-reported long-term persisting symptoms in Ecuador.

Dear Editor and reviewers, thank you very much for your effort in observing our manuscript and offering us some comments intended to improve our manuscript. 

We have completed a full revision which includes answers to all your comments and suggestions. All changes are highlighted in red within the main manuscript and this point-by-point letter.

Reviewer 1

The article by Izquierdo-Condoy et al. is the result of a lot of work. Based on an electronic questionnaire they evaluated post covid conditions in 1003 patients without Long Covid (LC) and 1100 patient who claimed to have at least one symptom defining Long Covid.  Their main and original conclusions are that “High altitude resident were more likely to report persisting symptoms versus those living at lower altitude and that the highest proportion of persisting symptoms was observed among those who received an incomplete vaccine scheme.”

Dear reviewer, thanks so much for your insightful comments, We have changed the entire conclusions section

These conclusions are not supported by the facts. There is a confusion all along the manuscript between the number of patients with LC /No LC and the number of symptoms of LC which makes difficult interpretation of data and support overinterpretation made by the authors 

Thanks again, We have clarified the results section and highlighted all the changes

When the variable studied is patient as reported in table 1, no difference is noted between patient living at high altitude versus low altitude. This is obviously confirmed in the risk analysis in table 5 where altitude is not a risk for LC in these patients.

In the symptomatologic analysis it is reported that among the 1100 patients that report LC “at least” 7746 events were collected and grouped into 52 symptoms. Among these symptoms’ mood changes, tachycardia, decrease libido, insomnia palpitation is reported more frequently in patient living at high altitude. This analysis should first be compared to NO long covid cohort as these features may be due to altitude alone. This univariate analysis should be also confirmed by multiple regression analysis as some of these factors may be linked such as mood, insomnia.

Dear reviewer, we appreciate your comment, please review our response. The total number of participants who took part in this investigation was 2103, of whom 1100 reported LC. According to the analysis proposed by reviewer 1, we consider that this would not have been possible. Although all respondents (n=2103) suffered from COVID-19, in those who responded not having developed LC the process of filling out the survey ended automatically. For this reason, it would be impossible to compare LC symptoms with a cohort that did not have LC.

 Moreover, Red Maca is a well know plant medication in South America reported as efficient on mood, decrease libido … (https://doi.org/10.3390/ph9030049 ) and was proposed as a treatment for COVID-19 and such medication might be a confounding factor.   Any way, if these symptoms are confirmed to be different between patient living in high and low altitude the conclusion proposed by the authors is not supported by the data and should be changed.

We appreciate your suggestion about the probable effect of "Red Maca" on the symptoms self-reported by our participants. We conducted a literature search based on your suggestion, however, we found no scientific information or other sources that would warrant the consumption of Red Maca in Ecuador from the onset of the COVID-19 pandemic (March 2020). For these reasons, we do not believe it prudent to grant a confounding effect to Red Maca, at least in the findings reported but we acknowledge dthis in the limitations as well as in the references section.

On vaccine it is more confusing. If the question is:  does vaccination protect from Long covid? the analysis should compare cohorts (LC and No LC) for the different vaccination scheme and period.

Here the question seems; is there any relationship between symptoms in long covid patient and vaccination scheme? In table 3 is reported the distribution of symptoms according to vaccination scheme in long covid cohort. The lecture of table 3 is ; hallucinations, confusion, loss of muscle strength, burning sensation in any part of the body, shaking chills, thirst and vomit are more frequently reported in LC patient receiving a complete vaccine scheme and a booster. This is interesting but this does not mean that vaccine is protective of LC. What is happening what does this mean? To solve the question a Multi Component Analysis should clarified the possible association between symptoms and vaccine scheme.

As mentioned before, data on vaccination are not available for respondents who reported not having LC, nevertheless, We added a multi-component analysis to clarify the effects of the number of vaccine doses and boosters on the reported LC symptoms.   

Terminology

Please homogenize the terms “Long covid” “Post-acute symptoms” and “Post COVID”… all along the manuscript

Several other terms have been used to describe prolonged symptoms following COVID-19 illness, such as "long COVID," "postacute sequelae of SARS-CoV-2 infection (PASC)," "postacute COVID-19," "chronic COVID-19," and "post-COVID syndrome" [9-13]. Despite the creation of case definitions, there are no widely accepted clinical diagnostic criteria for "long COVID" [14]. However, as of October 1, 2021, there is a new International Classification of Diseases, Tenth Revision, Clinical Modification (ICD-10) for unspecified post-COVID conditions, which is U09.9, which was approved by the CDC.

We replaced all terms referring to sequels with "Long-COVID". 

Specific comments

Abstract: Please rewrite appropriately

We have corrected the abstract.

Page 2 line 63:  sinus?

We have corrected this word

Page 6 Table 2: there are to many data in the manuscript, and I suggest to put the table 2 in supplementary data file

We changed table 2 as supplementary file 1.

Page 9 line 196 you cannot “identified at least …. “Some events. Please precise how many events you identified.

We have added precise information on the number of events identified.

Page 13 the end of table 3 does not fit the upper part and it is difficult to get the total of each column. 

We corrected the data layout at the end of the table.

In figure 3;4 and 5 it is not clear for me if axis of the orders are distribution of events or symptoms. Please clarify. The lecture is that the number or reported symptoms in patient with LC is lower when vaccinated for at least two doses.

The ordinate axis addresses the number of LC symptoms, we give this axis title to facilitate the understanding of the figures.

However, participant receiving the vaccine after their COVID had more symptoms (53.7%). This suggests a comment. Does this is a social desirability bias (post covid vaccine and deleterious effect) or this is realty “true” and need complementary investigation.

We clarify this information in the manuscript.

Conclusion

This is an interesting work that merit to be published. However, there is a serious need to clarify the research questions.

Analysis of the number and distribution of symptoms is interesting, but very weak in terms of reliability and interpretation, especially in the situation of LC which is a disease mostly with subjective claims, many bias (including social desirability, insurance needs) ….. The strength of the paper is the good and precise description of symptoms on a large cohort of patient, which is in fact not so frequently reported allowing to complete the current knowledge on this difficult public health problem.

We clarify the description of the research results. We address social desirability bias in the study limitations section. 

I would add the role of vaccination scheme and the time before and after natural infection on the Post covid condition (at least one claim/no claim at all) in the 2103 patients. As only 3 patients in the LC cohort have received less than two doses (99.9% having received at least 2 doses) and hypothesizing that it is approximately the same in Non LC cohort, it seems likely that no difference will appear, but need to be investigated as the repartition of the vaccine scheme before/after may differ in the two group

We developed and added an additional analysis to clarify the effect of vaccines on the LC.

Reviewer 2 Report

The novelty of the work is very less or virtually it has no novelty at this time point. The author should extensively revise the study with some novel data with correlation. % of female are more prone to the disease, they had fatigue etc. are well established at.

The authors may correlate their data with the environmental or socio economic or educational status of the participants. Such data would provide a new dimension to the study.

Title: It needs a  drastic shortening. What is long COVID?

It could be

 Epidemiological Analysis of COVID-19 condition in

 Ecuador

Abstract: The very first paragraph is wrongly written.

It could be

 Some patients who recovered from COVID-19 have experienced a range of persistent or new clinical symptoms.

From the abstract, it seems that already known reports are only studied in the country Ecuador.

Divide your intro part to 3 sections

·         *Current scenario of COVID-19.

·         * COVID 19 in your country

·         * The lacuna and your hypothesis.

In this section some of the relevant info form https://pubmed.ncbi.nlm.nih.gov/34390474/ and https://pubmed.ncbi.nlm.nih.gov/32982622/ and  https://pubmed.ncbi.nlm.nih.gov/32339832/ be extracted. After reading the above articles the author may able to speak better in their article or collect some data form internet (environmental?) and go ahead to discuss their study.

In material methods

Describe inclusion and exclusion criteria.

As I have mentioned, please discuss the results as per the above commented way.

Author Response

Reviewer 2

The novelty of the work is very less or virtually it has no novelty at this time point. The author should extensively revise the study with some novel data with correlation. % of female are more prone to the disease, they had fatigue etc. are well established at.

Dear reviewer, we appreciate your comments, nevertheless, we still, believe our results add some new insights to the vast scientific knowledge around covid-19, especially since we have analyzed our results by altitude. In order to fulfill your suggestions, we added a multiclass correlation analysis on the effects of vaccines with LC symptoms.

The authors may correlate their data with the environmental or socio economic or educational status of the participants. Such data would provide a new dimension to the study.

According to the certainties provided by the instrument, ethnicity as part of the socioeconomic context and altitude as part of the environmental context were studied. Since this is an organic condition, we believe that the correlation of LC with other confounders is difficult to explain. But we have acknowledged this within the limitation section

Title: It needs a  drastic shortening. What is long COVID?

It could be

 Epidemiological Analysis of COVID-19 condition in  Ecuador

We have changed the tittle accordingly to your suggestion

Abstract: The very first paragraph is wrongly written.

It could be

 Some patients who recovered from COVID-19 have experienced a range of persistent or new clinical symptoms.

We have corrected the entire abstract and all the typos were amended.

From the abstract, it seems that already known reports are only studied in the country Ecuador.

We have clarified this information.

Divide your intro part to 3 sections 

  • *Current scenario of COVID-19.
  • * COVID 19 in your country
  • * The lacuna and your hypothesis.

 In this section some of the relevant info form https://pubmed.ncbi.nlm.nih.gov/34390474/ and https://pubmed.ncbi.nlm.nih.gov/32982622/ and  https://pubmed.ncbi.nlm.nih.gov/32339832/ be extracted. After reading the above articles the author may able to speak better in their article or collect some data form internet (environmental?) and go ahead to discuss their study.

We restructured this section and added additional information as requested

In material methods

Describe inclusion and exclusion criteria.

We have added this information.

As I have mentioned, please discuss the results as per the above commented way.

We have restructured the entire discussion and reviewed the document entirely, many thanks for your suggestions

Round 2

Reviewer 1 Report

Anwers to reviewer requires have been appropriately provided

Author Response

Dear Reviewer

Thanks so much for your input,  the comments to the minor observations made by the editor were amended and they are included in the current version.

Many thanks

Reviewer 2 Report

It seems the authors have addressed all comments. 

Author Response

Dear Reviewer, Thanks so much for your endorsement and the minor comments from the editor.

please find attached the current version with the changes highlighted in red

Many Thanks

Esteban Ortiz